# Validation of the Achievement Emotions Questionnaire for Experimental Science Education (AEQ-S)

**DOI:** 10.3390/bs12120480

**Published:** 2022-11-26

**Authors:** Kevin Macías León, M. Ángeles de las Heras Pérez, Raquel Romero Fernández, Yolanda González Castanedo, Pedro Sáenz-López

**Affiliations:** Department of Integrated Didactics, Faculty of Education, Psychology and CC Sport, University of Huelva, 21071 Huelva, Spain

**Keywords:** emotions, experimental sciences, psychometric properties, adolescents

## Abstract

The Achievement Emotions Questionnaire (AEQ), based on the control-value theory of achievement emotions, has been used in many fields of knowledge and has been translated into many languages. The main objective of this study was to adapt and validate it for the experimental sciences. A sample of 491 participants aged between 11 and 17 years (*M* = 13.73, *SD* = 1.19) from secondary schools in different localities in the provinces of Huelva and Lanzarote, Spain was used. The results obtained from the various statistical analyses showed that the questionnaire was valid and reliable. The main contribution of the work presented is to broaden the field of application of a test on emotions, AEQ-S, to experimental sciences. The AEQ-ES will allow us to learn the emotional profiles of students, thus providing information to teachers, and will be very useful for future research aimed at the study of emotions in experimental sciences.

## 1. Introduction

It is quite common to find a significant number of students with a certain degree of dissatisfaction and lack of motivation in classrooms. These students often display boredom in the classroom, refuse to attend classes, want to leave school as soon as possible, and some even find no meaning in the work they do [1]. In a study by [2] on the causes of boredom in the classroom, it was observed that the main cause identified by students was the uselessness of some subjects, their poor contextualisation and the methodology used by teachers, but they also indicated the demeanour of the teachers. According to [3], emotional experiences are always present and meaningful. Emotions are very important in the teaching process [4], influencing students’ motivation and academic achievements, as they may feel excited while studying and be proud of their achievements, or they may have negative emotions, stemming from poor grades or boredom during lessons [5].

It is evident and already more than demonstrated [6] that the emotions of the teacher are transmitted to the students and that these are the main factor in the generation of the classroom climate. One study, based on the *crossover* theory formulated by [7], which consists of the belief that emotions can be triggered directly or indirectly from the emotions of others, goes on to analyse how the teacher’s emotions trigger emotions in students. Their own emotions play an important role and should not be neglected in the day-to-day teaching profession. Hence the importance of teachers learning to regulate their emotions [8] and becoming emotionally intellectual [9], given that they are the main emotional leaders of their students. According to [10] (p. 5), “the teacher’s ability to capture, understand and regulate the emotions of his or her students is the best index of the emotional balance of his or her class.” Teachers need to be aware of and manage the emotions that students experience in class, and in this sense, teachers who use boring teaching methods or produce classes with many failed students will bring on negative emotions such as anguish, indignation [11] and boredom [2]. According to [12], these negative emotions are associated with a deterioration in students’ attention, increasing demotivation. Conversely, there are positive emotions that typically generate beneficial effects on the construct of learning [13] and are related to a variety of adaptive school behaviours, such as engagement and learning [14,15], as they favour inductive reasoning and creativity [16] and widen the focus of attention, leading to greater exploration of the environment, more creative responses and novel reflections [17].

With increasing evidence, and more and more authors defend the characterisation of current teaching as an emotional practice, in which affective and cognitive processes are involved, and also maintain that emotions are essential in the learning process [18,19]. Studies such as that of [20] have already shown that children’s and adolescents’ emotions are related to their academic performance. Positive emotions such as enjoyment of learning generate positive links to achievement, and negative emotions such as exam anxiety demonstrate negative links. In the same way, some studies report that positive emotions facilitate learning, contribute to academic achievement [21] and predict satisfaction with life [22]. In contrast, although some of them achieve the extrinsic motivation of learners to a fair extent [23], negative emotions generally have a negative effect on learning [24] and have no effect on life satisfaction [22].

It is also beginning to be acknowledged that many strategies to improve well-being are aimed at developing positive emotions as a way of achieving higher levels of well-being [25]. Psychological well-being has been approached from different theories, among which the self-determination theory stands out, in which psychological functioning is based on positive and healthy experiences, which depend on the personal characteristics of each individual [26]. Psychological well-being encompasses life satisfaction, which [27] is defined as the degree to which a person evaluates the overall quality of his or her life as a whole in a positive way. Due to the importance of the construct, numerous instruments for its measurement have been developed over recent decades. However, the most notable is the Satisfaction with Life Scale [28]. Its application in educational contexts shows how higher levels demonstrate better social relations, educational attainment and physical health [29]. In addition, they demonstrate more favourable attitudes towards teachers and school [30], as well as higher academic engagement [31] and academic aspirations [32]. If we focus on the case of science teaching and learning, according to [33] attribution theory, pupils generate attitudes and emotions towards science during their time at school depending on their successes or failures. It is striking, as [34] point out, that emotions towards science change negatively with increasing age, with negative emotions being more prominent in secondary education than in primary education. This issue has led more and more researchers to focus on the influence of emotions in science education [18], with some, such as [35], focusing on addressing problems faced by students, including disengagement and negative attitudes towards science education and also the unsatisfactory approach with which science education is delivered [36].

Ultimately, as has been shown, academic environments trigger a wide variety of different emotions: students enjoy learning something new, feel proud of their results, feel annoyed by excessive homework or get bored during a lesson. However, the measurement of these achievement emotions or emotions that are directly linked to achievement activities is very recent [37], especially due to the lack of instruments for this purpose [38]. This gap is filled by the introduction of the Emotional Achievement Questionnaire (AEQ) [39]. According to [38], the AEQ is a measurement instrument that allows for the assessment of various achievement emotions in different settings. This instrument is based on the control-value theory of achievement emotions (CVTAE) [40], which highlights the importance of students’ emotions in achievement motivation by placing emotions as the central and proximate principle in explaining students’ behaviour and cognition [41].

The AEQ has been translated and adapted for several subjects [42,43,44,45,46], and even validated for the purpose of assessing emotions in physics classes [47]. Recently, [11] proposed exploring achievement emotions in physical education and [48] adapted and validated the AEQ for the Spanish context, given the satisfactory results obtained in other contexts and subjects.

For all these reasons, and due to the great need to change achievement emotions towards experimental sciences in students, as has been shown above, we propose adapting and validating the AEQ for experimental sciences in secondary school, in order to create an instrument that highlights emotional needs in the teaching and learning of science.

## 2. Methods

### 2.1. Participants

The sample consisted of 491 high school students (1st ESO–4th ESO) aged 11–17 years (*M* = 13.73, *SD* = 1.19), of whom 239 (47.5%) were girls, 233 (48.7%) were boys and 19 (3.9%) preferred not to say. Nonrandom sampling was carried out [49] following the criteria given by MacCallum et al. (2001) [50] for establishing the sample size, who suggest a conservative radius of 20 participants for each scale item to be assessed, in order to minimise errors. In this case, a minimum of 480 participants was required. The students belonged to 5 secondary schools located in Huelva and Lanzarote (Spain).

### 2.2. Instruments

#### 2.2.1. Achievement Emotions Questionnaire for Science (AEQ-S)

The AEQ-S (Appendix A) was adapted from the AEQ-PE [48], which was itself adapted from the Achievement Emotions in Preadolescent Students Questionnaire (AEQ-PA) [42]. This instrument consists of 24 items divided into six emotions (four for each emotion). These emotions are: boredom (e.g., “I’m bored of the target class”), hopelessness (e.g., “Even before I enter the target class, I know I won’t do well”), anger (e.g., “I feel angry after the target class”), anxiety (e.g., “I feel nervous in the target class”), enjoyment (e.g., “I’m glad it’s worthwhile to go to the target class”) and pride (e.g., “I’m proud of my participation in the target class”).

The responses are rated on a Likert-type scale that ranges from 1 (strongly disagree) to 5 (strongly agree).

#### 2.2.2. Life Satisfaction Scale

The Diener Life Satisfaction Scale [28] consists of five items that assess life satisfaction through people’s overall judgement of their life. For the present work, we used the Spanish translation of [51], which presents the items as follows: (a) “In most aspects, my life is as I want it to be,” (b) “So far, I have got from life the things I consider important,” (c) “I am satisfied with my life,” (d) “If I could live my life over again, I would repeat it as it has been” and (e) “The circumstances of my life are good.” In this version of the instrument, the number of response options was reduced (from 7 in the original version) so that the values range from 1 to 5, where 1 is “strongly disagree” and 5 is “strongly agree.” This instrument is a single-factor scale and has been used in a large number of studies, displaying very good psychometric properties.

### 2.3. Procedure

Taking into account the objective of the study, a bibliographic search was carried out to find specific investigations to evaluate aspects of the emotional involvement of students. Finally, we focus on the Value-Control Theory of Achievement Emotions (CVTAE). Specifically, taking this last validation and the main quadrants of the CVTAE into account, the emotions considered were pride, anxiety, enjoyment, pride, boredom and hopelessness.

The next step was to carry out the adaptation of this scale. The version for students from the area of Physical Education (AEQ-PE) was adapted to the context of natural sciences. The questionnaires were prepared and contact made with the head teachers of the secondary schools chosen for the study, explaining all the contents of the study and the way in which it would be carried out. Subsequently, family consent was sought. In this case, the sampling technique was convenience, non-probability sampling.

Before participants filled in the questionnaires, a pilot sample of 30 students from Grade 8 was tested. None expressed comprehension problems with the items. Teachers were also informed of the need for each participant to mark their questionnaire, as after 30 days, these participants would complete another questionnaire in order to check temporal stability.

Finally, data from all students who volunteered to participate in the study were collected during school hours and digitised.

### 2.4. Data Analysis

The reliability and validity of the questionnaire was analysed through a study of its psychometric properties. Thus, the scale was subjected to an analysis in which its internal consistency and temporal stability were studied. In turn, the factor structure of the scale was confirmed by a confirmatory factor analysis, using the most currently recommended fit indices to evaluate the proposed models [52]: χ^2^/df, comparative fit index (CFI), Tucker–Lewis index (TLI), goodness-of-fit index (GFI), incremental fit index (IFI), root mean square error of approximation (RMSEA) and standardised root mean square residual (SRMR). Values equal to or higher than 0.90 for CFI, TLI, GFI and IFI, lower than 3 for χ^2^/df and lower than 0.06 for RMSEA and SRMR are considered good-fit indices [53,54]. Finally, to test the performance of the scale in comparison to other scales, and more specifically whether achievement emotions (AEQ or independent variable) predict life satisfaction (SWL or dependent variable), a regression analysis was carried out.

The different analyses of the study were carried out using SPSS 24.0 and AMOS 24.0 statistical software (IBM, Armonk, NY, USA).

## 3. Results

### 3.1. Confirmatory Factor Analysis

The results of the confirmatory factor analysis using the maximum likelihood method showed a good fit with the model (Figure 1): χ^2^ (237) = 590, *p* < 0.001, χ^2^/df = 2.491, CFI = 0.934, GFI = 0.909, IFI = 0.935, TLI = 0.924, RMSEA = 0.055, SRMR = 0.055. The factor weights between items were statistically significant (*p* < 0.001) and ranged from 0.48 to 0.87.

### 3.2. Assessment of Internal Consistency and Temporal Stability

The results obtained from Cronbach’s alpha showed adequate internal consistency of the AEQ-S (Table 1). Temporal stability was measured in 49 participants with 30 days of difference between both measurements. Next, convergent validity was tested, as all correlations were statistically significant (*p* < 0.001), showing negative relationships between positive emotions (enjoyment and pride) and negative emotions (hopelessness, anxiety, boredom and anger), with the highest correlation coefficients being obtained in the relationships between enjoyment and boredom (−0.76) and between enjoyment and pride (0.81).

### 3.3. Criterion Validity Analysis

To measure the relationship of the AEQ-S dimensions with other variables, a regression analysis was carried out. This analysis aimed to predict the outcome of the dependent variable, life satisfaction, through the independent variable or achievement emotions. The results showed (Table 2) how positive achievement emotions predict life satisfaction, in contrast to negative achievement emotions. This is corroborated by the higher regression weight of positive emotions, which explain approximately 35% of life well-being, and are also significant (*p* < 0.001).

## 4. Discussion

The objective in this manuscript was to validate the AEQ for experimental science lessons in high school. For this purpose, the AEQ-PE [48] was adapted for this subject. This scale was for various reasons. Firstly, this questionnaire is focused on a subject (physical education) at the same academic level. Secondly, the AEQ-PE values some of the most important emotions that were found to be the most important in the CVTAE [3] (hopelessness, enjoyment, anxiety, boredom, pride and anger). Thus, the Achievement Emotions Questionnaire for Experimental Science (AEQ-S) was developed. The psychometric properties of this scale were measured through different statistical data, such as confirmatory factor analysis, temporal stability and internal consistency.

Confirmatory factor analysis revealed a model structured in six dimensions, as achieved in [47] and [55]. Furthermore, the fit indices were also adequate and somewhat better than those shown by the AEQ-ES (primary education) in the Chinese [56] and Italian [57] versions. The explanation for this may be due to the age of the participants, as they were younger than in this study, and therefore there is a difference in cognitive maturity and the ability to distinguish between emotions. The factor weights shown between the items (0.48–0.87) are in harmony with other validations of this scale, such as the one for preadolescent students (AEQ-PA) [42] or the one validated in physical education (AEQ-PE) [48]. These results show that there was coherence in the participants’ responses and thus that they were not randomly answered. Although the anxiety variable showed lower factor loadings than the other variables, very similar results have been found in studies such as those of [42] and [58].

For internal consistency, temporal stability [59] and Cronbach’s alpha were acceptable, as was the case in other research to date [44,60], although the values were slightly lower (0.67–0.86) than in [39] (0.77–0.93), indicating that students were able to understand the items for the most part, as their responses were quite consistent within each variable or dimension, and their responses were also consistent over time. The anxiety variable, which showed very high internal consistency and a low, although valid, value for temporal stability, should be highlighted.

Taking into account the theoretical framework of this research, which states that positive emotions promote satisfaction with life and negative emotions do not, and to establish the external validity of the scale, a regression analysis was carried out, with emotions as predictors of life satisfaction. The results showed moderate percentages of explained variation, which were higher for positive emotions (enjoyment 11% and pride 15%) than for negative emotions (hopelessness 6.35%, anxiety 5.2%, boredom 3.3% and anger 4.5%). Thus, the analysis of the regression weights suggests that the generation of positive emotions and the avoidance of negative ones during science lessons will favour students’ life satisfaction. In this respect, the results are partially in line with those obtained by [24], as they highlighted that positive emotions predict increases in life satisfaction, while negative emotions have weak effects and do not interfere with the benefits of positive emotions. Consequently, since the correlations established were significant and coherent with the theoretical framework, the external validity of the scale was evidenced in the sciences.

Descriptive studies show higher perceptions in the positive emotions than in the negative ones, e.g., data shared by several studies [47,61], including the Turkish adaptation of the Achievement Emotions Questionnaire for Mathematics (AEQ-M) [62]. This shows that students demonstrate more positive emotions towards experimental science, although one study found that German students showed higher levels of anger (negative emotion) [45]. The internal validity of the AEQ-ES is supported by the pattern of correlations obtained between all variables, offering logical and adequate relationships, as positive (hopelessness, anxiety, boredom and anger), results in line with several previous studies [12,38,44,62]. Again, anxiety showed lower values in the correlations with the variables, indicating emotions (pride and enjoyment) were positively correlated, and negative correlations with negative ones where the participants encountered some kind of difficulty in relation to the items linked to the anxiety variable, probably due to a lack of understanding, since according to previous studies, anxiety shows negative effects [44]. However, all correlations were statistically significant. These results allow us to understand that students are able to differentiate between emotions in different school situations and provide psychometric support for the AEQ-S. However, some limitations can be considered. Firstly, reference should be made to the variables used, in this case only six, and the two remaining positive emotions (relief and relaxation) that were not incorporated. However, the version selected for adaptation did not make use of these either. Secondly, although the sample was not small (491), for these cases in which the aim is to validate some kind of scale, larger samples are usually selected in order to obtain even more consistent results. In addition, participants were selected from only two provinces. Therefore, in future research, it would be very appropriate to carry out the study with a larger number of provinces and to be able to expand the study sample.

## 5. Conclusions and Future Directions

The importance of emotions in the educational context is more than evident, especially in the field of experimental sciences, and thus there is a need for research to obtain more information on this subject, in order to provide innovations that lead to an improvement in the teaching and learning process. This is why in this study the validation of a scale that fulfils this objective was carried out. The confirmatory factor analysis, internal consistency, temporal stability and relationship with other scales provided satisfactory results, thus supporting the reliability of the Achievement Emotions Questionnaire for Experimental Sciences and allowing its validity to be confirmed. This leads us to the conclusion that this questionnaire fulfils its function correctly, although with some nuances related to anxiety, where some difficulties arose. As a future line of research and thanks to this questionnaire, it will be possible to learn the different emotional profiles shown by students in experimental science classes, allow teachers to adapt their methodology to the emotional requirements of these students, and, furthermore, it will be very useful for the world of research, as future studies focusing on emotions in relation to the teaching and learning of experimental science can be carried out. As a limitation to this study, it should be taken into account that the application of this scale was validated for students aged 12–16 years (high school) and should not be used for students of different ages.

## Figures and Tables

**Figure 1 behavsci-12-00480-f001:**
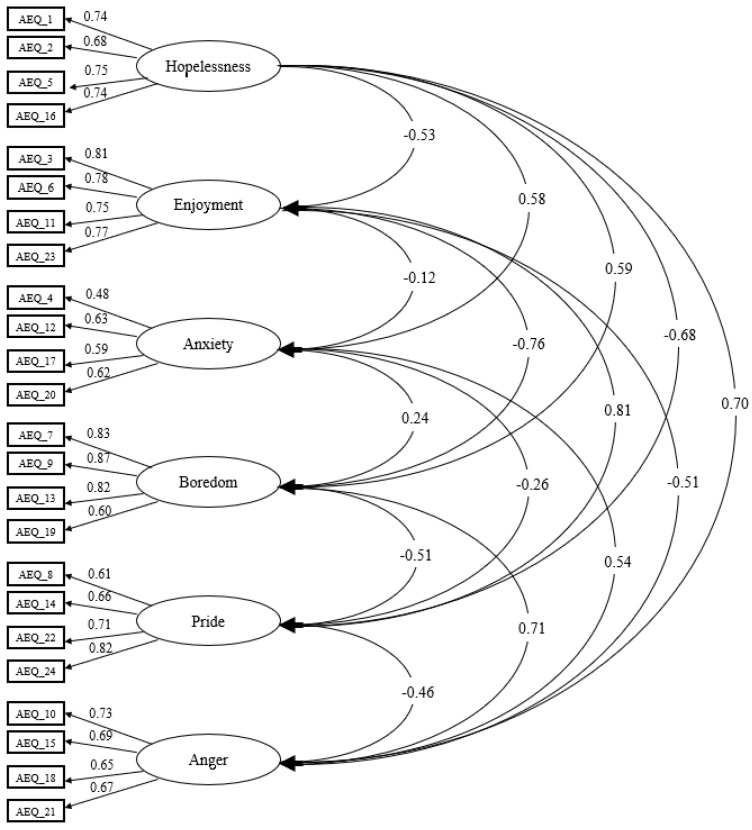
Confirmatory factor analysis of the Achievement Emotions Questionnaire for Experimental Science (AEQ-S). The ellipses represent the factors and the rectangles represent the specific elements.

**Table 1 behavsci-12-00480-t001:** Descriptive statistics, internal consistency, temporal stability, and convergent validity.

Emotions	*M*	*S* *D*	*α*	TS	1	2	3	4	5	6
Hopelessness	1.79	0.93	0.82	0.80	-	−0.53	0.58	0.59	−0.68	0.70
Enjoyment	3.65	0.97	0.86	0.82		-	−0.12	−0.76	0.81	−0.51
Anxiety	2.38	0.91	0.67	0.49			-	0.24	−0.26	0.54
Boredom	2.23	1.04	0.86	0.81				-	−0.51	0.71
Pride	3.77	0.95	0.80	0.85					-	−0.46
Anger	1.67	0.83	0.77	0.76						-

Note. *M* = mean; *SD* = standard deviation; *α* = Cronbach’s alpha; TS = temporal stability. All correlations were significant (*p* < 0.001).

**Table 2 behavsci-12-00480-t002:** Regression analysis results.

Variables	R^2^	B	T	*p*
**SWL**	0.063			
**Hopelessness**		−0.238	−5.7	0.000
**SWL**	0.114			
**Enjoyment**		0.306	7.91	0.000
**SWL**	0.052			
**Anxiety**		−0.219	−5.161	0.000
**SWL**	0.033			
**Boredom**		−0.155	−4.097	0.000
**SWL**	0.147			
**Pride**		0.358	9.18	0.000
**SWL**	0.045			
**Anger**		−0.225	−4.78	0.000

## Data Availability

The data are available to anyone. They can communicate directly with the corresponding author.

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
