# Peer review of "Validation of the Achievement Emotions Questionnaire for Experimental Science Education (AEQ-S)"

_behavsci, 2022, doi:10.3390/bs12120480_

Round 1

Reviewer 1 Report

This work focuses on an emerging theme in the field of education and psychology, such as the topic of emotions. To incorporate it as a research variable, it is necessary to have valid and reliable measurement instruments for the different curricular areas. The main contribution of the article is to expand the field of application of a previous test to experimental sciences.

The work is well structured, although it needs to be improved in some minor aspects, as indicated in the attached file.

On the other hand, some parts of the text have been underlined in order the authors review the language used.

Author Response

Thanking you for your review and your observations, which undoubtedly improve the submitted manuscript, we show below the response to your observations that you can also find in the attached manuscript.

REVIEW

COMMENT

CHANGE

Page 1, líne 29 of the review manuscript

teacher’s manner

the way of being of the teachers

Page 1, líne 34 of the review manuscript

It is evident and, has been more than demonstrated [6], that the teacher's emotions are passed on to students, and are the main factor in the generation of classroom climate.

It is evident and it is already more than demonstrated [6], that the emotions of the teacher are transmitted to the students and that these are the main factor in the generation of the classroom climate.

Page 2, line 49 of the review manuscript

,

The comma indicated in the text is removed

Page 2, líne 60 of the review manuscript

In the same vein

in the same way

Page 3, líne 130 of the review manuscript

was this scale added to the AEQ-S test?

It was not added, it was lost to demonstrate the external validity of the AEQ-S scale

Page 3, líne 145,146 of the review manuscript

They request to delete some comments that are repeated in the introduction.

The following paragraph is deleted:

In addition, it was noted that the Achievement Emotions Questionnaire (AEQ), 145 based on this theory, has been used for several subjects and translated into several languages 146 and was recently validated for physical education [48]

Page 3, líne 156 of the review manuscript

 2nd ESO

Grade 8

Page 5, líne 185 of the review manuscript

Translate the emotions of the figure into English

They are translated as can be seen in the text

Página 6, línea 212

reduce font size in the table

The type of tables in the document is reduced and homogenized

Página 6, Tabla 2

Pasar de R2 a R2

Se cambia

Página 7, línea 223

These results demonstrate that there was consistency between variables in the participants' responses

These results show that there was coherence in the participants´responses

En REFERENCIAS

Follow APA 7 edition standards and remove the city of edition from the books

It is done in all book references as can be seen in the manuscript

Página 10, línea 351

ent, 2017, 88(5),

2017, 88(5)

Reviewer 2 Report

1. Abstract should contain a short description of the research contribution.

2. The current Introduction does not show the need for questionnaire development for experimental sciences.

3. Please add a description of the sampling technique in Procedure.

4. Please add pretest results in Methods.

5. Please standardize the format of the table design.

6. Discussion should focus on the similarities and differences between the research results and the literature. 

7. I suggest to add the headings of contribution, suggestion, and research limitation in Conclusions.

Author Response

Comments and Suggestions for Authors

Thanking you for your review and your observations, which undoubtedly improve the submitted manuscript, we show below the response to your observations that you can also find in the attached manuscript.

  1. Abstract should contain a short description of the research contribution.

The main contribution of the work presented is to broaden the field of application of a test on emotions, AEQ-S, to experimental sciences (this comment is included in the page 1, line 17 and 18 of the attached manuscript).

  1. The current Introduction does not show the need for questionnaire development for experimental sciences.

In the introduction, there is general talk of taking emotions into account in the classroom, there is talk of the distancing of students from science subjects and the influence of emotions in science teaching, so there is a need to know them in order to act on them and improve them, since it is scientifically known that the increase in positive emotions improves performance. The instrument that is validated is to know the emotions of the students towards the experimental sciences . (these comments are included in the page 1, line 78 and 86 of the attached manuscript and following pages)

  1. Please add a description of the sampling technique in Procedure.

It is included in the procedure that the sampling technique was for convenience, non-probability sampling (Page 3, line 153 and 154).

  1. Please add pretest results in Methods.

The previous test in the investigation was carried out with the sole purpose of knowing if the students understood what they were being asked in order to redo those items that were not understandable by them. As detailed in the manuscript, there were no problems for the students to understand it, so it was not necessary to modify it. That is why there are no results as such from this initial test. (these comments are included in the page 3, line 154 and 155 of the attached manuscript).

  1. Please standardize the format of the table design.

The type of table, the font type and size of the tables have been standardized and homogenized as can be seen in the manuscript

  1. Discussion should focus on the similarities and differences between the research results and the literature. 

The reviewer's observation is attended to, since the discussion is carried out taking into account the results of other investigations.

  1. I suggest to add the headings of contribution, suggestion, and research limitation in Conclusions.

At the reviewer's request, section 5 of conclusions has been transformed into the conclusions and future lines section. In this section, in addition to presenting the conclusions of the study, the future lines of research that derive from it and the limitations of the study have been added.

Round 2

Reviewer 2 Report

Dear Authors,

You have revised the manuscript and has improved significantly, so I will suggest editor that it can be accepted this revision.

Best regards

Jian-Hong Ye